# Improving Ordinal Conformal Prediction by Stepwise Adaptive Posterior Alignment

## Abstract

Ordinal classification (OC) is widely used in real-world applications to categorize instances into ordered discrete classes. In risk-sensitive scenarios, ordinal conformal prediction (OCP) is used to obtain a small contiguous prediction set containing ground-truth labels with a desired coverage guarantee. However, OC models often fail to accurately model the posterior distribution, which harms the prediction set obtained by OCP. Therefore, we introduce a new method called *Adaptive Posterior Alignment Step-by-Step* (APASS), which reduces the distribution discrepancy to improve the downstream OCP performance. It is designed as a versatile, plug-and-play solution that is easily integrated into any OC model before OCP. APASS first employs an attention-based estimator to adaptively estimate the variance of the posterior distribution using the information in the calibration set, then utilizes a stepwise temperature scaling algorithm to align the posterior variance predicted by OC models to the better variance estimation. Extensive evaluations on 10 real-world datasets demonstrate that APASS consistently boosts the OCP performance of 5 popular OC models.

## 1 Introduction

Ordinal classification (OC) (Diaz & Marathe, 2019; Gao et al., 2017; Geng, 2016; Guo et al., 2008; Can Malli et al., 2016; Huo et al., 2016; Wen et al., 2020) plays a crucial role in high-stakes domains like healthcare (Liu et al., 2019) and finance (Manthoulis et al., 2020) by categorizing instances into ordered discrete classes. Robust uncertainty quantification is critical beyond accurate point predictions to avoid costly or dangerous outcomes caused by prediction errors. To this end, various methods have been developed for estimating predictive uncertainty in deep neural networks, such as confidence calibration (Guo et al., 2017), MC-Dropout (Gal & Ghahramani, 2016), and Bayesian neural networks (Smith, 2013), but they lack formal guarantees. Conformal Prediction (CP) (Vovk et al., 1999; 2005; Lei et al., 2018; Wen et al., 2020; Romano et al., 2020; Angelopoulos & Bates, 2021; Angelopoulos et al., 2021) addresses this gap by providing a distribution-free, post-processing approach that generates prediction sets (PS) guaranteed to contain the true label with a specified coverage probability, which generally design non-conformity scores to quantify the deviation the degree between the model's predictive outcomes and the data distribution.

Recent works on OC demonstrate substantial benefits of assuming the underlying conditional distribution to be unimodal for OC tasks (Diaz & Marathe, 2019; Gao et al., 2017; Guha et al., 2024; Belharbi et al., 2019; Cardoso et al., 2023). Some rely on label smoothing methods, which convert one-hot target labels into unimodal prior distributions to be used as the reference for the training loss. Some works learn a non-parametric unimodal distribution as a constraint optimization problem in the loss function. In the unimodal context of ordinal classification, Ordinal Conformal Prediction (OCP) (Lu et al., 2022; Xu et al., 2023) is designed to generate contiguous prediction sets using the posterior distribution predicted by OC models. In contrast to Adaptive Prediction Sets (APS), which calculate the scores by accumulating the sorted softmax probabilities in descending order, Ordinal-APS calculates the score by accumulating softmax probabilities of the contiguous prediction set with the minimum set size. However, the existing OCP methods neglect the possible variance misalignment of the OC models, which leads to inefficient PS.

In this work, we empirically observe the variance misalignment between the predicted posterior distribution and the oracle posterior distribution in a synthetic dataset. Specifically, a noticeable

reduction in the size of PS is observed when we align the predicted posterior to the oracle posterior. Further, our theoretical analysis supports the empirical findings by demonstrating a decrease in the upper bound of the prediction set size as the predicted posterior approaches the oracle posterior.

Inspired by our analytical findings, we introduce the *Adaptive Posterior Alignment Step-by-Step* (**APASS**), which serves as a *plug-and-play* component that can be integrated into any OCP framework to enhance its performance. The method consists of two key parts: **1**) We introduce an *attention*-based estimator that adaptively estimates the variance misalignment of an input sample by examining similar samples in the calibration set; **2**) a stepwise alignment algorithm that optimizes the calibrate the variance misalignment. The stepwise method can gradually amend the variance misalignment and produce more compact prediction sets.

To evaluate the effectiveness of APASS, we conduct extensive empirical assessments on real-world benchmarks, showing that APASS consistently improves the performance of OCP on 10 real-world datasets by 14.2% on average with 5 typical ordinal classification methods. The unstable performance of non-stepwise alignment baselines highlights the superiority of consistent improvement.

The contributions of this paper are summarized as follows:

- We identify the variance misalignment issue in current OC models that the existing OCP method neglects and theoretically prove that ignoring the misalignment will harm the efficiency of PSs in the context of OCP.

- We introduce the *Adaptive Posterior Alignment Step-by-Step* (APASS) method, a stepwise approach designed to reduce the PS size by reducing the distribution discrepancy using posterior variance alignment.

- We conduct extensive evaluations to show that APASS consistently improves the existing OCP method on various OC models. Specifically, the empirical results show the superiority of stepwise design to one-step baselines.

## 2 BACKGROUND

### 2.1 ORDINAL CLASSIFICATION.

In this study, we explore ordinal classification, which assigns labels to input instances based on a naturally ordered set of classes. We define the input space as $\mathcal{X} \subset \mathbb{R}^d$ and the ordered set of classes as $\mathcal{Y} = \{1, 2, \ldots, K\}$. The primary objective is to accurately predict the class label of input data using an OC model, denoted as $\hat{f} : \mathcal{X} \to \mathbb{R}^K$. Consider a scenario where the random variables $X$ and $Y$ are drawn from the combined space $\mathcal{X} \times \mathcal{Y}$ under a joint distribution $P_{X,Y}$. It is assumed that the true conditional distribution $P_{Y|X}$ is *unimodal*. This implies that for any given input instance $x \in \mathcal{X}$, the probability distribution $P(Y = y|X = x)$ peaks at a certain class $y$. The prediction of our model, therefore, hinges on $\hat{y} = \arg\max_{y \in \mathcal{Y}} \hat{p}_y(x)$, where $\hat{p}(y|x) = \text{softmax}(\hat{f}_\theta(y|x))$ represents the estimated probability that the input $x$ corresponds to class $y$.

### 2.2 ORDINAL CONFORMAL PREDICTION.

Ordinal Conformal Prediction (OCP) leverages the output of ordinal classifiers, symbolized by $\hat{p}(x)$, to construct a function $\mathcal{C} : \mathcal{X} \to 2^{\mathcal{Y}}$. This function maps input instances to a set of potential classes, ensuring a specific, user-defined confidence level. As a distribution-free methodology, OCP generates reliable prediction sets without making assumptions about the underlying data distribution. Formally, consider the following setup: 1) A **calibration set** comprising $n$ i.i.d. data points $\{(X_i, Y_i)\}_{i=1}^n$. These data points differ from the training data used to develop the ordinal classifier. 2) A new test instance $X_{n+1} \in \mathcal{X}$ and a target variable $Y_{n+1} \in \mathcal{Y}$. The primary objective is to construct a **prediction set** $\mathcal{C}_{n,1-\alpha}(X_{n+1})$ that remains minimal yet while ensuring that it satisfies **marginal coverage** at the confidence level $1 - \alpha$:

$$\mathbb{P}\Big(Y_{n+1} \in \mathcal{C}_{n,1-\alpha}(X_{n+1})\Big) \geq 1 - \alpha, \tag{1}$$

Furthermore, the prediction set should provide **conditional coverage** at the same confidence level:

$$\mathbb{P}\Big(Y_{n+1} \in \mathcal{C}_{n,1-\alpha}(X_{n+1})|X_{n+1} = x\Big) \geq 1 - \alpha, \quad \forall x \in \mathcal{X}. \tag{2}$$

**Oracle Prediction Set.** In ideal settings, accessing the oracle conditional distribution, denoted as $p(y|x)$, we construct an optimal prediction set satisfying Eq (2). It is formalized by:

$$\mathcal{C}_{1-\alpha}^{\text{oracle}}(x) = \Big[l_{1-\alpha}^{\text{oracle}}(x), u_{1-\alpha}^{\text{oracle}}(x)\Big], \tag{3}$$

where for any confidence level $\tau \in (0, 1]$, the boundaries of the prediction set are calculated as:

$$\Big(l_{\tau}^{\text{oracle}}(x), u_{\tau}^{\text{oracle}}(x)\Big) := \underset{(l,u)\in\mathbb{R}^2 : l \leq u}{\arg\min} \left\{ u - l : \sum_{j=l}^{u} p(y^j|x) \geq \tau \right\}. \tag{4}$$

**Practical Solution.** In practice, direct access to the Oracle conditional distribution, $p(y|x)$, is often infeasible. Instead, the trained ordinal classifier is utilized to approximate this function, which we denote as $\hat{p}(y|x)$. Among various OCP methods that utilize $\hat{p}(y|x)$ to determine prediction sets, our research adopts the Ordinal-APS method (Lu et al., 2022). This approach integrates CP techniques to generate contiguous prediction sets using a calibrated threshold $\hat{\tau}$ to meet the desired coverage at the level of $1 - \alpha$:

$$\hat{\mathcal{C}}_{n,1-\alpha}(x) = \Big[\hat{l}_{\hat{\tau}}(x), \hat{u}_{\hat{\tau}}(x)\Big]. \tag{5}$$

However, the estimated distribution $\hat{p}(y|x)$ often exhibits discrepancies, such as variance misalignment, compared to the oracle. Current OCP methods neglect these discrepancies, which can affect the efficiency of the PSs generated by the OCP method.

## 2.3 PROBABILITY CALIBRATION

Probability calibration, also known as confidence calibration (Guo et al., 2017), aims to ensure that the softmax probabilities predicted by neural networks accurately reflect the actual probabilities of correctness. To measure the degree of miscalibration, the Expected Calibration Error (ECE) is commonly used, quantifying the discrepancy between accuracy and confidence. ECE partitions the predictions into $M$ equally spaced bins and calculates a weighted average of the difference between the accuracy and confidence within each bin. Formally, ECE is defined as:

$$\text{ECE} = \sum_{m=1}^{M} \frac{|B_m|}{n} |\text{acc}(B_m) - \text{conf}(B_m)|, \tag{6}$$

where $\text{acc}(\cdot)$ and $\text{conf}(\cdot)$ denotes the average accuracy and confidence in bin $B_m$.

In prior works, it has been generally assumed that probability calibration improves the quality of conformal prediction sets (Angelopoulos et al., 2021; Gibbs et al., 2023). However, the specific impact of calibration methods on conformal prediction remains uncertain (Xi et al., 2024). Furthermore, existing calibration techniques are not explicitly tailored to OC models, which often assume unimodal distributions. The influence of these methods on OCP is thus still an open question.

## 3 RELATED WORK

**Ordinal Classification.** Recent works on ordinal classification demonstrate substantial benefits of assuming the underlying conditional distribution to be unimodal. Label smoothing methods convert one-hot target labels into unimodal prior distributions to be used as the reference for the training loss. SORD (Diaz & Marathe, 2019) constructs the ground-truth probability distribution using linear exponentially decaying distributions based on a metric loss function $\ell(y^t, y^i)$ that penalizes the distance between the actual value $y^t$ and the $i$-th prediction value $y^i$. This method uses the cross-entropy loss to train a neural network model. DLDL (Gao et al., 2017), on the other hand, constructs the probability of the $i$-th prediction using a normal probability density function and minimizes the Kullback-Leibler divergence between the predicted probability distribution and the ground-truth

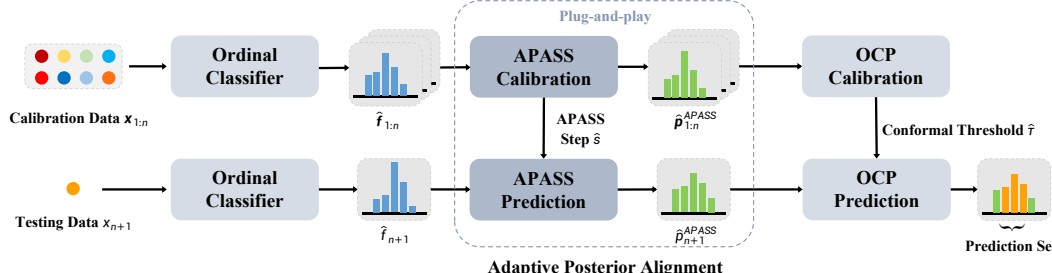

Figure 1: **Overview of APASS.** APASS is a plug-and-play module that adjusts the conditional distribution as predicted by ordinal classifiers before applying the OCP approach. This versatility allows it to be integrated seamlessly with various ordinal classifiers and OCP methods. Details of **APASS-Calibration** and **APASS-Prediction** are presented in Algorithm 1 & 2.

labels. R2CCP (Guha et al., 2024) proposes a loss function similar to label smoothing losses, which penalizes the probability based on the distance between the actual value $y^t$ and the $i$-th prediction value $y^i$ and uses a Shannon entropy regularizer to prevent the density estimator from collapsing to a Dirac distribution. But these methods are often sub-optimal since the assumed priors might not reflect the true distribution, classes might not be equispaced categories, and additionally, test predictions might not necessarily be unimodal. Some other methods (Belharbi et al., 2019; Cardoso et al., 2023) learn a non-parametric unimodal distribution as a constraint optimization problem in the loss function, which is not only difficult to optimize but also does not guarantee unimodality on testing data.

**Conformal Prediction**  Conformal prediction is a statistical framework characterized by a finite-sample coverage guarantee. One of the main goals of CP methods is to generate a compact prediction set. APS (Romano et al., 2020) introduces techniques aimed at achieving coverage that is similar across regions of feature space. RAPS (Angelopoulos et al., 2020) presents a regularized version of APS for Imagenet. The first work proposing the CP method for ordinal classification is Ordinal-APS (Lu et al., 2022), and another work proposes a similar approach in the context of ordinal conformal risk control (Xu et al., 2023).

The primary focal points of CP are reducing prediction set size and enhancing coverage rate. COPOC (Dey et al., 2023) proposes a special neural network model to ensure the ordinal classifier outputs an unimodal distribution and thus reduces the size of the ordinal prediction set size. In contrast, our model-agnostic method can be applied to different ordinal classifiers. Closely related to our insight, some works also utilize the information in the calibration set to generate compact prediction sets NCP (Ghosh et al., 2023) proposes to use non-parametric nearest neighbors for calibration, and in the context of sequential data, HopCPT (Auer et al., 2024) uses Modern Hopfield Networks to model the data similarity, and reweight the non-conformity score based on the similarity.

**Probability Calibration**  Guo et al. (2017) investigates the problem of confidence calibration in modern neural networks and finds post-processing methods like TS can effectively calibrate predictions. Standard TS improves average calibration but reduces confidence for all predictions. AdaTS (Joy et al., 2023) predicts a different temperature value for each input, allowing it to selectively increase or decrease confidence as needed. Esaki et al. (2024) proposes an accuracy-preserving calibration method using the Concrete distribution as a probabilistic model on the probability simplex, which outperforms previous TS methods in accuracy-preserving calibration tasks.

## 4 METHOD

In this section, we *empirically* and *theoretically* analyze the influence of distribution discrepancies on the efficiency of PSs generated by the OCP method: PSs will be smaller if distribution discrepancies are smaller. Therefore, we propose a *plug-and-play* method to optimize the PS efficiency, named **A**daptive **P**osterior **A**lignment **S**tep-by-**S**tep (**APASS**). APASS first employs a variance estimator (attention-based or kNN-based, see Section 4.2) to estimate the distribution discrepancies using data

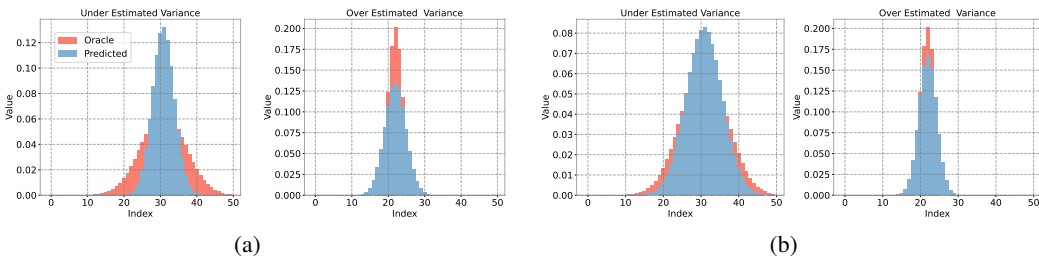

(a)                                                                          (b)

Figure 2: Empirical Evidence for Variance Alignment. (a) Variance discrepancy between oracle and predicted distribution. (b) Distributions after variance alignment using temperature scaling

in the calibration set. Then, to make use of the estimated variance misalignment, we proposed a stepwise method that gradually adjusts the predicted posterior using temperature scaling (TS) with a small step size to ensure a monotonic decrease in PS size in Section 4.3.

### 4.1 MOTIVATION

Recent works on OC demonstrate substantial benefits of assuming the underlying conditional distribution to be unimodal for OC tasks. To encourage unimodality, label-smoothing methods convert one-hot target labels into unimodal prior distributions to be used as the reference for the training loss, and some other methods use non-parametric unimodal distribution as a constraint in the loss function. However, these methods are optimized for accuracy, not posterior distribution discrepancy.

To investigate the impact of distribution discrepancies between the prediction posterior distribution, $\hat{p}(y|x)$, and the oracle distribution, $p(y|x)$. We first generated a *heteroscedastic* synthetic dataset, then trained an OC model in this dataset. This allows us to access the Oracle distribution $p(y|x)$ and better mirror real-world data scenarios. Figure 2 (a) highlights the existing discrepancies: the OC models tend to overestimate the variance when the actual variance is low and underestimate it when the actual variance is high. Moreover, the oracle prediction set has an average size of 8.54, contrasting with 9.62 for the set derived from $\hat{p}(y|x)$. This considerable difference highlights a critical variance misalignment problem, which compromises the efficiency of the prediction set size.

Inspired by the probability calibration method, we then employ temperature scaling to align the prediction distribution to the oracle distribution. Specifically, we use a grid-search strategy to minimize the discrepancy between the variance of the predicted posterior distribution and the variance of the oracle posterior distribution. In Figure 2 (b), we illustrate the aligned distribution. The results show a more closely aligned distribution, with the average size of the prediction set reduced to 8.84. These findings demonstrate that temperature scaling not only facilitates variance alignment but also enhances the efficiency of the prediction set, effectively resolving issues of variance misalignment. Then, we establish a theoretical explanation for our empirical finding, which explains how distribution discrepancy affects the relationship between the estimated prediction set, $\hat{\mathcal{C}}_{n,1-\alpha}(X_{n+1})$, and the oracle prediction set, $\mathcal{C}_{1-\alpha}^{\text{oracle}}(X_{n+1})$. We begin by defining some necessary assumptions:

**Assumption 1** (i.i.d. data). *The data $\{(X_i, Y_i)\}_{i=1}^{n+1}$ are i.i.d. from some unknown joint distribution.*

**Assumption 2** (Unimodality). *For any $x \in \mathbb{R}^m$, the conditional distribution of $Y|X = x$ is unimodal; i.e. there exists $y^0 \in \mathbb{R}$ (depending on $x$), such that $p(y^0 + y''|x) \leq p(y^0 + y'|x)$ if $y'' \geq y' \geq 0$, and $p(y^0 + y''|x) \leq p(y^0 + y'|x)$ if $y'' \leq y' \leq 0$.*

**Assumption 3** ($\eta$-inconsistency). *Let $F(y|x)$ denote the cumulative distribution function of $Y|X = x$, and define $\hat{F}(y|x)$ as the estimate of cumulative distribution function, i.e., $\hat{F}(y^j|x) := \sum_{i=1}^{j} \hat{p}_\theta(y^i|x)$. Then, we assume for all $j \in 1, \ldots, K$,*

$$\mathbb{P}\left[\sup_{j \in \{1, \ldots, K\}} \left[|\hat{F}(y^j|X) - F(y^j|X)|\right] \leq \eta\right] \geq 1 - \eta. \tag{7}$$

**Assumption 4** (Regularity). *For any $x \in \mathbb{R}^m$ and $j \in \{1, 2, \ldots, K\}$, $1/H < p(y^j|x) < 2/H$, for some $H > 0$.*

---

**Algorithm 1** APASS Calibration

---

**Input:** logits of calibration data $\hat{\boldsymbol{f}}_{1:n} = \hat{f}_\theta(\boldsymbol{x}_{1:n})$ and variance estimator $\widehat{\text{Var}}_\psi(Y|X, \mathcal{D}_{calib})$
**Output:** aligned posterior $\hat{\boldsymbol{p}}_{1:n}^{\text{APASS}}$ and the TS steps $\hat{s}$

1: Calculate $t(\boldsymbol{x}_{1:n})$ by Eq 9, 10, 12      ▷ distribution discrepancy on the calibration set
2: $\hat{\boldsymbol{p}}_{1:n} \leftarrow \text{softmax}(\hat{\boldsymbol{f}}_{1:n})$
3: $|\hat{\mathcal{C}}_{n,1-\alpha}(\boldsymbol{x}_{1:n})| \leftarrow \text{OCP-Calibration}(\hat{\boldsymbol{f}}_{1:n})$      ▷ run OCP-Calibration to get average PS size
4: $best \leftarrow |\hat{\mathcal{C}}_{n,1-\alpha}(\boldsymbol{x}_{1:n})|$      ▷ initialize best PS size
5: **for** $s \in \{1, \ldots, s_{max}\}$ **do**
6:      $\hat{\boldsymbol{f}}_{1:n} \leftarrow \hat{\boldsymbol{f}}_{1:n}/t(\boldsymbol{x}_{1:n})$      ▷ perform TS$(t(\boldsymbol{x}_{1:n}))$
7:      $|\hat{\mathcal{C}}_{n,1-\alpha}(\boldsymbol{x}_{1:n})| \leftarrow \text{OCP-Calibration}(\hat{\boldsymbol{f}}_{1:n})$      ▷ get new PS size
8:      **if** $|\hat{\mathcal{C}}_{n,1-\alpha}(\boldsymbol{x}_{1:n})| \leq best$ **then**      ▷ stop until PS size does not reduce
9:          $best \leftarrow |\hat{\mathcal{C}}_{n,1-\alpha}(\boldsymbol{x}_{1:n})|$
10:      **else**
11:          Break
12:      **end if**
13: **end for**
14: $\hat{\boldsymbol{p}}_{1:n}^{\text{APASS}} \leftarrow \text{softmax}(\hat{\boldsymbol{f}}_{1:n} * t(\boldsymbol{x}_{1:n})), \hat{s} \leftarrow s - 1$
15: **return** $\hat{\boldsymbol{p}}_{1:n}^{\text{APASS}}, \hat{s}$

---

Assumption 4 allows us to quantify the distribution discrepancy using $\eta$, where a higher $\eta$ value signifies a greater discrepancy. We leverage this quantification to theoretically analyze and establish an upper bound for $|\hat{\mathcal{C}}_{n,1-\alpha}(X_{n+1})|$:

**Theorem 1.** *For any $\alpha \in (0, 1]$, let $\hat{\mathcal{C}}_{n,1-\alpha}(X_{n+1})$ denote the prediction set at level $1 - \alpha$ for $Y_{n+1}$ obtained by applying OCP. The size prediction set $\hat{\mathcal{C}}_{n,1-\alpha}(X_{n+1})$ is bounded by the set of oracle prediction set $\mathcal{C}_{n,1-\alpha}^{oracle}(X_{n+1})$ as*

$$\mathbb{P}\left[\left|\hat{\mathcal{C}}_{n,1-\alpha}(X_{n+1})\right| \leq \left|\mathcal{C}_{n,1-\alpha}^{oracle}(X_{n+1})\right| + \gamma_n\right] \geq 1 - \xi_n, \tag{8}$$

*where $\gamma_n = 2 + H(3/n + 2\sqrt{(\log n)/n} + 5\eta)$ and $\xi_n = \eta + 2n^{-2}$.*

This theorem demonstrates that decreasing $\eta$, the measure of the discrepancy between estimated and actual distributions, results in a tighter upper bound for $|\hat{\mathcal{C}}_{n,1-\alpha}(X_{n+1})|$. Consequently, this reduction can effectively decrease the size of the prediction set $\hat{\mathcal{C}}_{n,1-\alpha}(X_{n+1})$. This provides theoretical guidance to find a practical way to enhance the efficiency of the PS by using probability calibration methods to reduce the distribution discrepancy. However, the straightforward application of probability calibration poses a challenge: we empirically find that probability calibration may harm the PS efficiency in the OCP setting (see Section 5.1). In the following sections, we address this challenge by introducing a novel stepwise distribution alignment method.

### 4.2 Measure Distribution Discrepancy with Posterior Variance Estimator

To optimize the PS efficiency by reducing distribution discrepancy, we need to measure it by comparing the posterior variance predicted by the OC model $\widehat{\text{Var}}_\theta(Y|X)$ and the posterior variance estimated using data from the calibration set $\widehat{\text{Var}}(Y|X, \mathcal{D}_{calib})$. It is straightforward to calculate the posterior variance predicted by the OC model:

$$\widehat{\text{Var}}_\theta(Y|X = x) = \sum_{j=1}^{K} \hat{p}(y^j|x) \cdot \left(y^j - \sum_{j=1}^{N} \hat{p}(y^j|x) \cdot y^j\right)^2. \tag{9}$$

Inspired from Auer et al. (2024), we estimate the posterior variance considering the calibration set using a weighted sum of prediction errors derived from calibration data. This sum prioritizes points similar to the test sample and aggregates their deviations. Specifically, we define the squared residual

---

**Algorithm 2** APASS Prediction

---

**Input:** logits of testing data $\hat{f}_{n+1} = \hat{f}_\theta(x_{n+1})$, variance estimator $\widehat{\mathrm{Var}}_\psi(Y|X, \mathcal{D}_{calib})$, TS steps $\hat{s}$
**Output:** logits after APASS $\hat{f}_{n+1}^{\mathrm{APASS}}$
  1: Calculate $t(x_{n+1})$ by Eq 9, 10, 12         ▷ distribution discrepancy of the testing sample
  2: **for** $s \in \{1, \ldots, \hat{s}\}$ **do**
  3:     $\hat{f}_{n+1} \leftarrow \hat{f}_{n+1}/t(x_{n+1}))$
  4: **end for**
  5: $\hat{f}_{n+1}^{\mathrm{APASS}} \leftarrow \mathrm{softmax}(\hat{f}_{n+1})$
  6: **return** $\hat{f}_{n+1}^{\mathrm{APASS}}$

---

error in the calibration set as $\epsilon_i{}^2 = (y_i - \hat{y}_i)^2$, For a new sample point $X = x$, the posterior variance considering the calibration set is estimated by:

$$\widehat{\mathrm{Var}}_\psi(Y|X = x, \mathcal{D}_{calib}) = \mathrm{softmax}\left(\beta\phi^T(x)\boldsymbol{W}_q^T\boldsymbol{W}_k\phi(\boldsymbol{x}_{1:n})\right)\boldsymbol{\epsilon}_{1:n}^2, \tag{10}$$

where $\boldsymbol{x}_{1:n}$ are samples' features and $\boldsymbol{\epsilon}_{1:n}^2$ are also squared residual errors in the calibration set, $\boldsymbol{W}_q$ and $\boldsymbol{W}_k$ are learned transformations applied before associating the query with the calibration data's keys, $\phi$ is an encoder, transforming raw features into appropriate representation vectors, and $\psi$ represents all model parameters. To effectively train the variance estimator, we utilize a leave-one-out (LOO) training strategy. The primary objective in the training phase is to minimize the mean squared error between the predicted squared errors $\hat{\boldsymbol{\epsilon}}_{1:n}^2$ and the actual squared errors $\boldsymbol{\epsilon}_{1:n}^2$. The training loss, therefore, is formulated as follows:

$$\mathcal{L}_\psi = \mathrm{MSE}\left(\widehat{\mathrm{Var}}_\psi(Y|\boldsymbol{x}_{1:n}, \mathcal{D}_{calib}), \boldsymbol{\epsilon}_{1:n}^2\right) \tag{11}$$

During the computation of $\widehat{\mathrm{Var}}_\psi(Y|x_i, \mathcal{D}_{calib})$, the weight of $\epsilon_i$ is masked as 0 to prevent leakage of actual error values into the model training process following LOO. With the two posterior variance estimators, we can finally define the distribution discrepancy of a sample $x$ as:

$$t(x) = \left(\frac{\widehat{\mathrm{Var}}_\psi(Y|X = x, \mathcal{D}_{calib})}{\widehat{\mathrm{Var}}_\theta(Y|X = x)}\right)^q \tag{12}$$

We then use $t(x)$ as the temperature in TS, which we refer to as $\mathrm{TS}(t)$. There is no variance discrepancy if $t(x) = 1$, which means no need to scale the distribution. If $t(x) > 1$, indicating the OC model underestimates the posterior variance, TS will make the distribution more even; and if $t(x) < 1$, indicating the OC model overestimates the posterior variance, TS will make the distribution more narrow. $q > 0$ is a hyper-parameter to determine the step size of TS: larger $q$ means a larger TS step given to same variance discrepancy.

### 4.3 Stepwise Posterior Alignment with Temperature Scaling

The next question is how to determine the step size $q$. A straightforward solution is to find the optimal $q^*$ by minimizing the ECE as other probability calibration methods do, named **A**daptive **P**osterior **A**lignment by **C**onfidence **C**alibration (**APACC**). However, we find this approach may harm the efficiency of PSs in practice. Therefore, inspired by gradient descent, we propose a sstepwisemethod that uses a small constant step size $q$ but looks for the optimal steps reducing the PSs' size.

Corresponding to CP methods, APASS comprises 2 components: APASS-Calibration (Algorithm 1) and APASS-Prediction (Algorithm 2), as illustrated in Figure 1. **APASS-Calibration** calibrates the posterior distribution of calibration data predicted by the OC model using distribution discrepancy measure (Eq. 12) step by step and outputs the steps of TS $\hat{s}$. The aligned distributions $\hat{\boldsymbol{p}}_{1:n}^{\mathrm{APASS}}$ are then fed to OCP-Calibration to calculate the non-conformity score and the conformal threshold $\hat{\tau}$. During test time, **APASS-Prediction** will first estimate the distribution discrepancy of a testing sample $x_{n+1}$ as $t(x_{n+1})$, then run $\mathrm{TS}(t(x_{n+1}))$ for $\hat{s}$ steps to get aligned posterior $\hat{p}_{n+1}^{\mathrm{APASS}}$, which OCP-Prediction will take to generate PS for the testing sample $x_{n+1}$.

Table 1: **Averaged results of $\Delta$Cov** comparing our method with baselines on 10 datasets and across 5 distinct OC models.

| *Dataset* | *Original* | *One-step method* | | *Stepwise method (Ours)* | |
|---|---|---|---|---|---|
| | | AdaTS | APACC-Att | APASS-kNN | APASS-Att |
| Breastcancer | $0.031_{\pm 0.054}$ | $0.028_{\pm 0.055}$ | $0.03_{\pm 0.051}$ | $0.033_{\pm 0.05}$ | $0.033_{\pm 0.051}$ |
| Community | $0.005_{\pm 0.029}$ | $0.005_{\pm 0.034}$ | $0.005_{\pm 0.029}$ | $0.005_{\pm 0.032}$ | $0.005_{\pm 0.031}$ |
| Concrete | $0.012_{\pm 0.031}$ | $0.012_{\pm 0.034}$ | $0.013_{\pm 0.033}$ | $0.011_{\pm 0.033}$ | $0.013_{\pm 0.032}$ |
| Diabetes | $0.025_{\pm 0.046}$ | $0.025_{\pm 0.055}$ | $0.023_{\pm 0.053}$ | $0.023_{\pm 0.052}$ | $0.026_{\pm 0.054}$ |
| Energy | $0.014_{\pm 0.036}$ | $0.013_{\pm 0.037}$ | $0.015_{\pm 0.036}$ | $0.014_{\pm 0.04}$ | $0.013_{\pm 0.041}$ |
| Forest | $0.011_{\pm 0.038}$ | $0.011_{\pm 0.036}$ | $0.011_{\pm 0.035}$ | $0.012_{\pm 0.037}$ | $0.012_{\pm 0.036}$ |
| Parkinsons | $0.002_{\pm 0.015}$ | $0.002_{\pm 0.015}$ | $0.002_{\pm 0.014}$ | $0.002_{\pm 0.015}$ | $0.002_{\pm 0.014}$ |
| Pendulum | $0.035_{\pm 0.045}$ | $0.037_{\pm 0.04}$ | $0.033_{\pm 0.039}$ | $0.037_{\pm 0.044}$ | $0.034_{\pm 0.042}$ |
| Solar | $0.017_{\pm 0.055}$ | $0.017_{\pm 0.054}$ | $0.018_{\pm 0.048}$ | $0.019_{\pm 0.047}$ | $0.018_{\pm 0.047}$ |
| Stock | $0.015_{\pm 0.065}$ | $0.014_{\pm 0.057}$ | $0.015_{\pm 0.06}$ | $0.014_{\pm 0.059}$ | $0.016_{\pm 0.054}$ |

## 5 EXPERIMENTS

This section presents the evaluation of APASS, designed to efficiently generate prediction sets with specified coverage for OC tasks. We tested APASS across diverse real-world datasets and established OC models, demonstrating consistent performance improvements over all baselines. An ablation study confirmed the essential contribution of each component, enhancing APASS's robustness and reducing its sensitivity to hyperparameter changes. The results affirm APASS's effectiveness and practicality in real-world applications.

**OC models.** To validate the effectiveness of APASS across different base OC models, we tested 5 popular OC models, including 3 label-smoothing methods and 2 methods that promote unimodality non-parametrically. Specifically, **SORD**, **DLDL**, **R2CCP** build different smoothed labels to encourage unimodal. **ELB** (Belharbi et al., 2019) enforces unimodality and label-order consistency via a set of non-parametric inequality constraints over all pairs of adjacent labels. **UN** (Cardoso et al., 2023) proposed a new neural network architecture that directly constrains the output to be unimodal.

**Baselines.** We compared our models against four baselines: 1) **Original:** The original Ordinal-APS model without adaptive calibration. 2) **APACC-Attn:** A one-step variant of our APASS method, which uses grid search to find the optimal step size $\hat{q}$ that minimizes ECE. This $\hat{q}$ is then applied to adjust the posterior during testing. 3) **AdaTS:** A state-of-the-art adaptive probability calibration method (Joy et al., 2023), which is incorporated into OC models before applying CP. 4) **APASS-kNN:** A variation that substitutes our attention-based variance measure with a distance-based alternative. Specifically, we use a kNN estimator to assess posterior variance Formally,

$$\widehat{\mathrm{Var}}_{kNN}(Y|X = x_{n+1}, \mathcal{D}_{calib}) = \frac{\sum_{i \in \mathrm{NN}(x_{n+1})} w_i^k |\hat{y}_i - y_i|}{\sum_{i \in \mathrm{NN}(x_{n+1})} w_i^k} + \frac{\min_{i \in \mathrm{NN}(x_{n+1})} d(x_i, x_{n+1})}{\max_{i,j \in \mathcal{D}_{calib}} d(x_i, x_j)} \widehat{\sigma} \quad (13)$$

where $\mathrm{NN}(x_{n+1})$ is the set of the nearest $k$ samples, $w_i = 1 - \frac{d(x_i, x_{n+1})}{\sum_{i \in \mathrm{NN}(x_{n+1})} d(x_i, x)}$ and $\widehat{\sigma} = \sqrt{\mathrm{Var}\left[\{y_i\}_{i \in \mathrm{NN}(x_{n+1})} \cup \{\hat{y}_{n+1}\}\right]}$. In our experiment, $k = 5$ and $d(\cdot, \cdot)$ is the Mahalanobis distance.

**Metrics.** The metrics used in our evaluation include $\Delta$**Cov**, defined as the absolute difference between the actual and target coverage given a specific target coverage level $1 - \alpha$, where smaller values indicate better validity and a value of zero implies perfect alignment. Additionally, we assess the average size of the prediction set, denoted as |**PS**|, to evaluate the efficiency of different methods.

**Datasets.** We evaluate our method using 10 popular real-world datasets to ensure the robustness and applicability of our approach. Specifically, these datasets include several from the 10 UCI Machine Learning Repository (Kolby et al., 2024). These datasets encompass a diverse range of domains and data characteristics.

Table 2: **Averaged results of |PS|** comparing our method with baselines on 10 datasets and across 5 distinct OC models. The last raw is the average percentage reduction. See Table 4 in the Appendix for complete results.

| Dataset | Original | One-step method | | Stepwise method (Ours) | |
|---|---|---|---|---|---|
| | | AdaTS | APACC-Att | APASS-kNN | APASS-Att |
| Breastcancer | $43.5_{\pm 3.5}$ | $40.6_{\pm 4.6}$(-6.8%) | $44.0_{\pm 3.8}$(+1.2%) | $41.3_{\pm 4.1}$(-5.0%) | $\mathbf{40.0}_{\pm 4.9}$**(-8.2%)** |
| Community | $21.2_{\pm 1.6}$ | $21.7_{\pm 1.8}$(+2.1%) | $23.9_{\pm 2.2}$(+12.7%) | $19.9_{\pm 1.8}$(-6.2%) | $\mathbf{19.1}_{\pm 1.7}$**(-9.9%)** |
| Concrete | $17.1_{\pm 1.7}$ | $17.1_{\pm 2.0}$(+0.2%) | $17.6_{\pm 1.7}$(+2.7%) | $16.0_{\pm 2.0}$(-6.6%) | $\mathbf{15.0}_{\pm 1.6}$**(-12.4%)** |
| Diabetes | $35.7_{\pm 3.6}$ | $35.8_{\pm 3.6}$(+0.0%) | $33.0_{\pm 3.7}$(-7.8%) | $34.1_{\pm 3.3}$(-4.7%) | $\mathbf{33.5}_{\pm 3.6}$**(-6.4%)** |
| Energy | $11.0_{\pm 0.3}$ | $11.1_{\pm 0.5}$(+0.7%) | $10.8_{\pm 0.2}$(-1.2%) | $9.1_{\pm 0.4}$(-17.1%) | $\mathbf{8.1}_{\pm 0.3}$**(-26.5%)** |
| Forest | $29.6_{\pm 2.8}$ | $29.2_{\pm 3.1}$(-1.3%) | $32.3_{\pm 3.1}$(+9.2%) | $27.6_{\pm 2.6}$(-6.7%) | $\mathbf{26.0}_{\pm 3.0}$**(-12.1%)** |
| Parkinsons | $9.4_{\pm 0.1}$ | $9.6_{\pm 0.6}$(+2.7%) | $9.0_{\pm -0.6}$(-3.9%) | $8.0_{\pm 0.4}$(-14.0%) | $\mathbf{6.8}_{\pm 0.1}$**(-27.9%)** |
| Pendulum | $10.4_{\pm 1.2}$ | $10.2_{\pm 0.6}$(-2.4%) | $11.0_{\pm 0.7}$(+5.8%) | $9.8_{\pm 0.7}$(-6.3%) | $\mathbf{9.5}_{\pm 1.0}$**(-9.0%)** |
| Solar | $17.4_{\pm 3.1}$ | $17.9_{\pm 3.2}$(+2.8%) | $24.6_{\pm 3.6}$(+41.2%) | $14.9_{\pm 3.6}$(-14.6%) | $\mathbf{13.4}_{\pm 3.9}$**(-23.2%)** |
| Stock | $13.2_{\pm 1.0}$ | $13.6_{\pm 0.9}$(+3.2%) | $12.9_{\pm 0.6}$(-1.8%) | $12.7_{\pm 0.4}$(-4.0%) | $\mathbf{12.3}_{\pm 1.0}$**(-6.4%)** |
| *Averaged Reduction* ↓ | | 0.12% | 5.81% | -8.52% | **-14.20%** |

**Evaluation Setup.** For each OCP method in one dataset, we randomly split the data into folds with 70% / 30% as $\mathcal{D}_{train}/\mathcal{D}_{\text{calib}} \cup \mathcal{D}_{\text{test}}$ for 3 times to train 3 different models. We conduct 10 random splits of calibration/testing sets for each OC model to estimate the empirical coverage and PS size. For all APASS experiments, we use the same step size $q = 0.05$

## 5.1 EMPIRICAL RESULTS

**Effectiveness of APASS across Multiple OC Models on Real-World Datasets.** We begin by comparing our method with several baselines on a diverse set of real-world datasets across five distinct OC models. As shown in Table 1, APASS-Att maintains a consistently low $\Delta$Cov value, indicating strong alignment between the predicted labels and the target labels. In terms of |PS|, as reported in Table 2, APASS-Att achieves significant reductions. On average, it reduces the size of the prediction sets by 14.20%, with reductions ranging from 6.4% to 27.9%. APASS-kNN using a less competitive non-parametric variance estimator brings an 8.52% reduction in average. Moreover, APASS-Att and APASS-kNN never increase the prediction set size compared to the original models, demonstrating their robustness and adaptability across various datasets and models.

**Superiority of Stepwise vs. One-Step Approaches.** As reported in Table 2, both stepwise methods, APASS-Attn and APASS-kNN, consistently outperform one-step baselines, including the state-of-the-art probability calibration method AdaTS and our one-step variant APACC-Att, which fail to reduce |PS| across all datasets. These one-step approaches fail to reduce the |PS| in 60% of cases. In contrast, the stepwise approach consistently reduces prediction set sizes, showcasing its superior ability to optimize prediction efficiency while maintaining strong model alignment. This advantage reinforces the value of the stepwise framework in real-world applications.

**Empirical Evidence of Synchronous Changes of PS Size.** One of the critical contributions of this work is the stepwise alignment approach, which uses the calibration set to determine the optimal TS steps. We validate this empirically by analyzing how prediction set sizes evolve during *stepwise temperature scaling*. Figure 3 demonstrates that prediction set sizes on both calibration and testing sets change synchronously across three OC models on the Community and Stock dataset. This validates that the stepwise method can accurately determine the TS steps based solely on the calibration set. The same synchronous behavior is consistently observed across other datasets, with comprehensive results included in Appendix B.

**Robust Performance of APASS and Computation Cost.** Hyperparameter sensitivity is crucial in real-world deployment, as hyperparameter selection typically requires human expertise and can be resource-intensive. Our empirical results demonstrate that APASS is largely insensitive to hyperparameter variation. Specifically, Table 3 shows the percentage reduction in |PS| when applying APASS with various step sizes $q$. While larger step sizes (e.g., 0.5 and 1.0) lead to

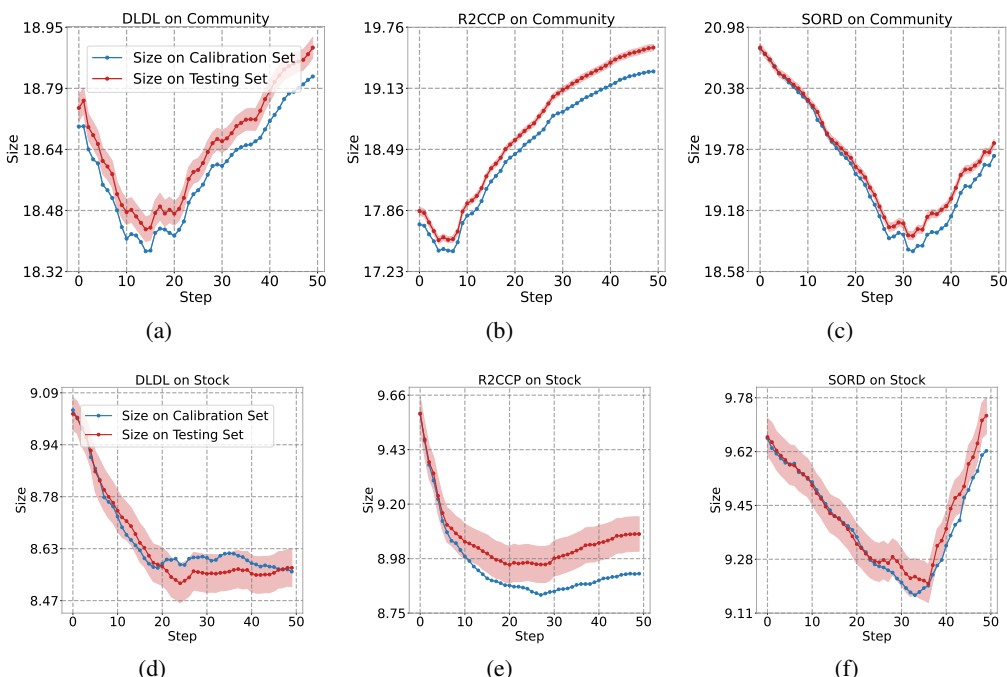

Figure 3: The sizes of the prediction set change synchronously on calibration and testing sets.

suboptimal results, using a step size below 0.1 consistently yields near-optimal outcomes. APASS is also computationally efficient. The posterior variance estimator and calibration training should be complete before deployment, so this part of the computation cost is not crucial. The following table illustrates that our method only costs about 9.2% computation overhead with $q = 0.05$ (our setting). However, if the step size $q$ is set to 0.01, the computation overhead will be 62% but only bring 0.1% extra reduction.

Table 3: Average size reduction after APASS-Att and computation cost in testing time with different step sizes $q$. The last raw is the result of the original Ordinal-APS.

| $q$ | 0.01 | 0.02 | 0.05 | 0.1 | 0.2 | 0.5 | 1.0 | / |
|---|---|---|---|---|---|---|---|---|
| *Averaged Reduction* | -14.3% | -14.2% | -14.2% | -13.8% | -8.7% | -5.3% | -1.7% | / |
| *Running time (s)* | 42.3 | 34.2 | 28.5 | 27.8 | 27.2 | 26.5 | 26.2 | 26.1 |
| *Overhead* | 62.1% | 31.0% | 9.2% | 6.5% | 4.2% | 1.5% | 0.4% | / |

## 6 CONCLUSION

In this paper, we find the issue of variance misalignment in popular ordinal classifiers, which will harm OCP. We empirically and theoretically show the efficiency of OCP can be improved if ordinal classifiers predict a more accurate conditional distribution. Thus, we introduce the APASS technique, which employs an attention-based variance estimator and stepwise temperature scaling to align the posterior variance modeled by ordinal classifiers with better variance estimation. Empirical evaluations on benchmark datasets demonstrated that APASS significantly enhances the performance of OCP methods without the need for hyperparameter tuning, offering a robust framework for high-stakes healthcare, finance, and beyond applications. **Limitation** of our methods is we only use variance to align the posterior, high-order moment such as skewness and kurtosis can be considered in future works.

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

## A    PROOF OF THEORY

The theoretical proof strikes the idea from Sesia & Romano (2021), which focuses on regression problems with continuous variables, whereas we concentrate on ordinal classification with discrete variables.

**Lemma 1.** *Def the event $\mathcal{A}$ as*

$$\mathcal{A} := \left\{ x : \sup_{j \in \{1,\ldots,K\}} |\hat{F}(y^j|x) - F(y^j|x)| > \eta \right\} \tag{14}$$

*Then, under Assumptions 1-3, for any $X \perp\!\!\!\perp \mathcal{D}^{train}$,*

$$\mathbb{P}[X \in \mathcal{A}] \leq \eta \tag{15}$$

*Furthermore, partitioning the calibration data points into*

$$\mathcal{D}^{\mathrm{cal},a} := \{i \in \{1,\ldots,n\} : X_i \in \mathcal{A}\}, \quad \mathcal{D}^{\mathrm{cal},b} := \{i \in \{1,\ldots,n\} : X_i \in \mathcal{A}^{\mathrm{c}}\} \tag{16}$$

*we have that, for any constant $c > 0$*

$$\mathbb{P}\left[|\mathcal{D}^{cal,a}| \geq n\eta + c\sqrt{n \log n}\right] \leq n^{-2c^2} \tag{17}$$

**Lemma 2.** *Under Assumptions 1–3, for any $\tau \in (0,1)$ and $X \perp\!\!\!\perp \mathcal{D}^{train}$,*

$$\mathbb{P}\left[|\hat{\mathcal{C}}_{n,\tau}(X)| \leq |\mathcal{C}_{n,\tau+2\eta}(X)| + 2\right] \geq 1 - \eta, \tag{18}$$

**Lemma 3.** *For any $\tau \in (0,1)$, let $\hat{Q}_\tau(E_i)$ denote the $\lceil \tau(n+1) \rceil$ smallest value among the conformity scores $\{E_i\}$ for $i \in \mathcal{D}^{cal}$, where $i \in \mathcal{D}^{cal}$ and*

$$E_i := \min\left\{\tau_t \in \{0, 1/T_n, \ldots, (T_n-1)/T_n, 1\} : Y_i \in \hat{\mathcal{C}}_{n,\tau_t}(X_i)\right\} \tag{19}$$

*Then, under Assumptions 1-3, for any $c > 0$,*

$$\mathbb{P}\left[\hat{Q}_\tau(E_i) \leq \tau + \epsilon_n\right] \geq 1 - 2n^{-2c^2}, \tag{20}$$

*where $\epsilon_n := 3/n + 3\eta + 2c\sqrt{(\log n)/n}$*

*Proof of Theorem 1.* Define $\epsilon_n := 3/n + 3\eta + 2c\sqrt{(\log n)/n}$ for any $c > 0$, as in Lemma 3. In the event that $\hat{Q}_{1-\alpha}(E_i) \leq 1 - \alpha + \epsilon_n$,

$$\begin{aligned}
\mathbb{P}&\left[|\hat{\mathcal{C}}_{n,\hat{Q}_{1-\alpha}(E_i)}(X)| \leq |\mathcal{C}_{n,1-\alpha+\epsilon_n+2\eta}(X)| + 2\right] \\
&\geq \mathbb{P}\left[|\hat{\mathcal{C}}_{n,1-\alpha+\epsilon_n}(X)| \leq |\mathcal{C}_{n,1-\alpha+\epsilon_n+2\eta}(X)| + 2\right] \\
&\geq 1 - \eta,
\end{aligned} \tag{21}$$

where the second inequality follows by applying Lemma 2 with $\tau = 1 - \alpha + \epsilon_n$. Further, as Lemma 3 tells us, the above event occurs with a high probability,

$$\mathbb{P}\left[\hat{Q}_{1-\alpha}(E_i) \leq 1 - \alpha + \epsilon_n\right] \geq 1 - 2n^{-2c^2} \tag{22}$$

in general, we have that

$$\mathbb{P}\left[|\hat{\mathcal{C}}_{n,\hat{Q}_{1-\alpha}(E_i)}(X)| \leq |\mathcal{C}_{n,1-\alpha+\epsilon_n+2\eta}(X)| + 2\right] \geq 1 - \eta - 2n^{-2c^2} \tag{23}$$

By Assumption 4, $p(y^j|x) > 1/H$ for all $j \in \{1, 2, \ldots, K\}$. This implies $\mathcal{C}_{n,\tau}(X)$ is $H$-Lipschitz as a function of $\tau$. Therefore,

$$\mathbb{P}\left[|\hat{\mathcal{C}}_{n,\hat{Q}_{1-\alpha}(E_i)}(X)| \leq |\mathcal{C}_{n,1-\alpha}(X)| + 2 + H(\epsilon_n + 2\eta)\right]$$

$$\geq \mathbb{P}\left[|\hat{\mathcal{C}}_{n,\hat{Q}_{1-\alpha}(E_i)}(X)| \leq |\mathcal{C}_{n,1-\alpha+\epsilon_n+2\eta}(X)| + 2\right] \quad (24)$$

$$\geq 1 - \eta - 2n^{-2c^2}.$$

Hence, setting $c = 1$ we have proved that

$$\mathbb{P}\left[|\hat{\mathcal{C}}_{n,\hat{Q}_{1-\alpha}(E_i)}(X)| \leq |\mathcal{C}_{n,1-\alpha}(X)| + \gamma_n\right] \geq 1 - \xi_n. \quad (25)$$

$\square$

*Proof of Lemma 1.* Inequation 15 and easily derived from definition of $\mathcal{A}$ in Eq. 14 and Assumption 3. As we know from the above that $\mathbb{P}[X \in A_n] \leq \eta$, for any $\epsilon > 0$, following Hoeffding's inequality,

$$\mathbb{P}\left[|\mathcal{D}^{\text{cal},a}| \geq n\eta + \epsilon\right] \leq \mathbb{P}\left[|\mathcal{D}^{\text{cal},a}| \geq n\mathbb{P}[X \in A_n] + \epsilon\right]$$

$$\leq \mathbb{P}\left[\frac{1}{n}\sum_{i=n+1}^{2n} \mathbb{1}[X_i \in A_n] \geq \mathbb{P}[X_i \in A_n] + \frac{\epsilon}{n}\right] \quad (26)$$

$$\leq \exp\left(-\frac{2\epsilon^2}{n}\right).$$

Therefore, setting $\epsilon = c\sqrt{n \log n}$, for some constant $c > 0$, yields

$$\mathbb{P}\left[|\mathcal{D}^{\text{cal},a}| \geq n\eta + c\sqrt{n \log n}\right] \leq n^{-2c^2}. \quad (27)$$

$\square$

*Proof of Lemma 2.* Consider the event $\mathcal{A}$ defined in Lemma 1. Let's consider the case where $X \in \mathcal{A}^c$. We can write $\hat{\mathcal{C}}_{n,\tau}(X) = [\hat{j}_1, \hat{j}_2]$ for some $\hat{j}_1, \hat{j}_2 \in \{1, \ldots, K\}$ such that $\hat{F}\left(y^{\hat{j}_2}\right) - \hat{F}\left(y^{\hat{j}_1 - 1}\right) \geq \tau$. Then, the triangle inequality implies $F\left(y^{\hat{j}_2}\right) - F\left(y^{\hat{j}_1 - 1}\right) \geq \tau - 2\eta$. Consider the oracle set $\mathcal{C}_{n,\tau+2\eta}(X)$, which we can write in short as $[l^*, u^*]$ for some $l^*, u^* \in \mathbb{R}$ such that $F(u^*) - F(l^*) \geq \tau + 2\eta$. Define $j_1', j_2' \in \{1, \ldots, K\}$ as the indices of the label immediately below and above $l^*, u^*$:

$$j_1' := \max\left\{j \in \{1, \ldots, m_n\} : y^j < l^*\right\}$$
$$j_2' := \min\left\{j \in \{1, \ldots, m_n\} : y^j > u^*\right\} \quad (28)$$

This definition implies

$$y^{j_2'} - y^{j_1'} \leq u^* - l^* + 2, \quad (29)$$

Furthermore,

$$\hat{F}\left(y^{j_2'}\right) - \hat{F}\left(y^{j_1'}\right) \geq \hat{F}(u^*) - \hat{F}(l^*)$$
$$\geq F(u^*) - F(l^*) - 2\eta \quad (30)$$
$$\geq \tau.$$

The result implies that $\hat{j}_2 - \hat{j}_1 \leq j_2' - j_1'$ because $\hat{j}_2 - \hat{j}_1$ is the minimal of $\hat{\mathcal{C}}_{n,\tau}(X)$. Then,

$$|\hat{\mathcal{C}}_{n,\tau}(X)| = y^{\hat{j}_2} - y^{\hat{j}_1} \leq y^{j_2'} - y^{j_1'}$$
$$\leq |\mathcal{C}_{n,\tau+2\eta}(X)| + 2 \quad (31)$$

if $X \in \mathcal{A}^c$. Finally, by applying Lemma 1,

$$\mathbb{P}\left[\left|\hat{\mathcal{C}}_{n,\tau}(X)\right| \leq |\mathcal{C}_{n,\tau+2\eta}(X)| + 2\right] = \mathbb{P}[X \in \mathcal{A}^c] \geq 1 - \eta \quad (32)$$

$\square$

*Proof of Lemma 3.* Take any $i \in \mathcal{D}^{\mathrm{cal},b}$, where $\mathcal{D}^{\mathrm{cal},b}$ is defined as in Lemma 1:

$$\mathcal{D}^{\mathrm{cal},b} := \{i \in \{1, \ldots, n\} : X_i \in A^{\mathrm{c}}\}, \tag{33}$$

For any fixed $t \in \{0, \ldots, n\}$ and $\tau_t = t/n$, omitting the explicit dependence on $X$ and $\hat{p}$, we can write $\hat{\mathcal{C}}_{n,\tau_t}(X) = \left[\hat{j}_1, \hat{j}_2\right]$, for some $\hat{j}_1, \hat{j}_2 \in \{1, \ldots, K\}$ such that $\hat{F}(y^{\hat{j}_2}) - \hat{F}(y^{\hat{j}_1-1}) \geq \tau_t$. Then

$$\begin{aligned}
\mathbb{P}\left[E_i \leq \tau_t\right] &= \mathbb{P}\left[Y_i \in \hat{\mathcal{C}}_{n,\tau_t}(X)\right] \\
&= F(y^{\hat{j}_2}) - F(y^{\hat{j}_1-1}) \\
&\geq \hat{F}(y^{\hat{j}_2}) - \hat{F}(y^{\hat{j}_1-1}) - 2\eta \\
&\geq \tau_t - 2\eta.
\end{aligned} \tag{34}$$

Above, the first inequality follows from the definition of $\mathcal{D}^{\mathrm{cal},b}$. Equivalently, we can rewrite this as

$$\mathbb{P}\left[E_i > \tau_t + 2\eta + \delta\right] \leq 1 - \tau_t - \delta, \tag{35}$$

for any $\delta > 0$. Now, partition $\mathcal{D}^{\mathrm{cal},b}$ into the following two disjoint subsets:

$$\begin{aligned}
\mathcal{D}^{\mathrm{cal},b1} &:= \left\{i \in \mathcal{D}^{\mathrm{cal},b} : E_i \leq \tau_t + 2\eta + \delta\right\} \\
\mathcal{D}^{\mathrm{cal},b2} &:= \left\{i \in \mathcal{D}^{\mathrm{cal},b} : E_i > \tau_t + 2\eta + \delta\right\}
\end{aligned} \tag{36}$$

We bound $|\mathcal{D}^{\mathrm{cal},b2}|$ with Hoeffding's inequality. For any $i \in \mathcal{D}^{\mathrm{cal}}$, define $\tilde{E}_i = E_i$ if $i \in \mathcal{D}^{\mathrm{cal},b}$ and $E_i = \tau_t$ otherwise. For any $\epsilon > 0$,

$$\begin{aligned}
&\mathbb{P}\left[|\mathcal{D}^{\mathrm{cal},b2}| \geq n(1 - \tau_t - \delta) + \epsilon\right] \\
&\leq \mathbb{P}\left[\frac{1}{n}\sum_{i \in \mathcal{D}^{\mathrm{cal},b}} \mathbb{1}\left[\tilde{E}_i > \tau_t + 2\eta + \delta\right] \geq \mathbb{P}\left[E_i > \tau_t + 2\eta + \delta\right] + \frac{\epsilon}{n}\right] \\
&= \mathbb{P}\left[\frac{1}{n}\sum_{i=1}^{n} \mathbb{1}\left[\tilde{E}_i > \tau_t + 2\eta + \delta\right] \geq \mathbb{P}\left[E_i > \tau_t + 2\eta + \delta\right] + \frac{\epsilon}{n}\right] \\
&\leq \mathbb{P}\left[\frac{1}{n}\sum_{i=1}^{n} \mathbb{1}\left[\tilde{E}_i > \tau_t + 2\eta + \delta\right] \geq \mathbb{P}\left[\tilde{E}_i > \tau_t + 2\eta + \delta\right] + \frac{\epsilon}{n}\right] \\
&\leq \exp\left(-\frac{2\epsilon^2}{n}\right)
\end{aligned} \tag{37}$$

Therefore, setting $\epsilon = c\sqrt{n \log n}$, for some constant $c > 0$, yields

$$\mathbb{P}\left[|\mathcal{D}^{\mathrm{cal},b2}| \geq n(1 - \tau_t - \delta) + c\sqrt{n \log n}\right] \leq n^{-2c^2} \tag{38}$$

As $|\mathcal{D}^{\mathrm{cal},b1}| = n - |\mathcal{D}^{\mathrm{cal},a}| - |\mathcal{D}^{\mathrm{cal},b2}|$, combining the above result with that of Lemma 1 yields:

$$\mathbb{P}\left[|\mathcal{D}^{\mathrm{cal},b1}| \geq n\tau_t + n\delta - n\eta - 2c\sqrt{n \log n}\right] \geq 1 - 2n^{-2c^2} \tag{39}$$

If we choose $\delta = \tau_t/n + \eta + 2c\sqrt{(\log n)/n}$

$$\mathbb{P}\left[|\mathcal{D}^{\mathrm{cal},b1}| \geq \tau_t(n+1)\right] \geq 1 - 2n^{-2c^2}, \tag{40}$$

which means

$$\mathbb{P}\left[\hat{Q}_{\tau_t}(E_i) \leq \tau_t + \tau_t/n + 3\eta + 2c\sqrt{(\log n)/n}\right] \geq 1 - 2n^{-2c^2} \tag{41}$$

Now, consider any continuous $\tau \in (0, 1]$, and $t' = \min \{t \in \{0, \ldots, T_n\} : \tau_t \geq \tau\}$. As $\tau_{t'} \geq \tau$, we know $\hat{Q}_{\tau_{t'}}(E_i) \geq \hat{Q}_\tau(E_i)$. Therefore,

$$
\begin{aligned}
\mathbb{P}\left[\hat{Q}_\tau(E_i) \leq \tau_{t'} + \tau_{t'}/n + 3\eta + 2c\sqrt{(\log n)/n}\right] \\
\geq \mathbb{P}\left[\hat{Q}_{\tau_{t'}}(E_i) \leq \tau_{t'} + \tau_{t'}/n + 3\eta + 2c\sqrt{(\log n)/n}\right] \\
\geq 1 - 2n^{-2c^2}.
\end{aligned}
\tag{42}
$$

As $\tau_{t'} = \tau + 1$. Therefore,

$$
\begin{aligned}
\mathbb{P}\left[\hat{Q}_\tau(E_i) \leq \tau + 1/n + \tau/n + 1/n^2 + 3\eta + 2c\sqrt{(\log n)/n}\right] \\
\geq 1 - 2n^{-2c^2}.
\end{aligned}
\tag{43}
$$

Finally, as $\tau \leq 1$ and $n \geq 1$, replacing $1/n + \tau/n + 1/n^2$ with $3/n$ will preserve the inequality and Lemma 3 is proved. □

## B EXPERIMENT

### B.1 SYNTHETIC DATASET

We employed a method involving multivariate normal distributions and linear combinations to generate synthetic data for our experiment. Initially, we created a random mean vector and a symmetric positive-definite covariance matrix to define the multivariate normal distribution. This distribution is used to generate a dataset of features. We computed the output means and variances by applying linear combinations of the generated features with randomly generated coefficients to produce output values. Precisely, the means are calculated as a linear combination of the features, while the variances are determined by squaring another linear combination of these features, ensuring non-negativity. Finally, output values are sampled from a normal distribution using the calculated means and variances, resulting in a comprehensive synthetic dataset for experimental analysis.

### B.2 ORDINAL CLASSIFIER

Vanilla cross-entropy loss ignores the ordinal relationship and non-uniform separation among labels. To learn better conditional mass function approximation $\hat{p}(y|x)$, many label smoothing methods convert one-hot target labels into unimodal prior distributions to be used as the reference for the training loss. Soft ordinal classification (SORD) constructs the ground-truth p.m.f. based on a metric loss function $\ell(y^t, y^i)$ that penalizes how far the true value $y^t$ is from the i-th prediction value $y^i$ (Diaz & Marathe, 2019):

$$
p(y^i) = \frac{e^{-\ell(y^t, y^i)}}{\sum_{k=1}^K e^{-\ell(y^t, y^k)}} \quad \forall y^i \in \mathcal{Y},
\tag{44}
$$

and use cross-entropy loss to train the neural network model. Deep Label Distribution Learning (DLDL) minimizes the KL divergence between the predicted probability and the ground-truth labels in a similar way:

$$
p(y^i) = \frac{\mathcal{K}(y^i|\mu, \sigma)}{\sum_{k=1}^K \mathcal{K}(y^k|\mu, \sigma)} \quad \forall y^i \in \mathcal{Y}.
\tag{45}
$$

where $\mathcal{K}(y^i|\mu, \sigma)$ is a normal p.d.f. The mean $\mu$ is set to the actual value, i.e., $\mu = y^t$, and $\sigma$ is usually determined by the data distribution. For the continuous label in the regression setting, Regression-to-Classification Conformal Prediction (R2CCP) converts regression into ordinal classification by discretizing the label space into $K$ bins. They proposed a loss similar to label smoothing losses with a Shannon entropy regularizer, which prevents the density estimator from collapsing to one-hot output (Guha et al., 2024):

$$
\mathcal{L}(\theta) = \sum_{k=1}^K \ell(y^t, y^k)\hat{p}_\theta(y^k|x) - \tau\mathcal{H}(\hat{p}_\theta(\cdot|x)).
\tag{46}
$$

Table 4: Full results of |PS| on 10 dataset with 5 OC models.

| Dataset | OC Model | Original | One-step method | | Stepwise method (Ours) | |
|---|---|---|---|---|---|---|
| | | | AdaTS | APACC-Att | APASS-kNN | APASS-Att |
| Breastcancer | DLDL | $42.53_{\pm4.36}$ | $45.3_{\pm4.62}$ | $43.13_{\pm2.5}$ | $39.35_{\pm2.97}$ | $\mathbf{36.35}_{\pm3.99}$ |
| | ELB | $41.9_{\pm3.74}$ | $30.17_{\pm6.02}$ | $42.48_{\pm4.95}$ | $41.83_{\pm4.12}$ | $\mathbf{41.7}_{\pm5.46}$ |
| | R2CCP | $43.0_{\pm2.05}$ | $49.61_{\pm4.76}$ | $43.95_{\pm3.95}$ | $41.03_{\pm4.68}$ | $\mathbf{39.96}_{\pm5.49}$ |
| | SORD | $43.77_{\pm3.72}$ | $28.93_{\pm2.73}$ | $43.38_{\pm2.47}$ | $40.08_{\pm4.31}$ | $\mathbf{39.05}_{\pm5.09}$ |
| | UN | $46.4_{\pm3.85}$ | $48.82_{\pm4.69}$ | $47.21_{\pm5.18}$ | $44.41_{\pm4.31}$ | $\mathbf{42.8}_{\pm4.59}$ |
| Community | DLDL | $18.77_{\pm1.45}$ | $19.54_{\pm2.11}$ | $21.79_{\pm2.73}$ | $18.51_{\pm1.23}$ | $\mathbf{18.4}_{\pm1.54}$ |
| | ELB | $16.4_{\pm1.56}$ | $17.94_{\pm1.14}$ | $17.83_{\pm2.1}$ | $16.22_{\pm2.06}$ | $\mathbf{16.0}_{\pm1.93}$ |
| | R2CCP | $17.98_{\pm1.26}$ | $18.73_{\pm1.97}$ | $22.81_{\pm2.41}$ | $17.55_{\pm1.62}$ | $\mathbf{17.35}_{\pm1.28}$ |
| | SORD | $20.9_{\pm1.85}$ | $18.33_{\pm1.72}$ | $20.98_{\pm1.46}$ | $19.6_{\pm2.48}$ | $\mathbf{19.27}_{\pm2.59}$ |
| | UN | $32.1_{\pm1.83}$ | $33.85_{\pm2.25}$ | $36.21_{\pm2.22}$ | $27.7_{\pm1.57}$ | $\mathbf{24.6}_{\pm1.38}$ |
| Concrete | DLDL | $10.31_{\pm1.96}$ | $9.26_{\pm1.5}$ | $10.66_{\pm-0.07}$ | $10.19_{\pm1.14}$ | $\mathbf{10.14}_{\pm1.92}$ |
| | ELB | $15.4_{\pm1.75}$ | $10.0_{\pm2.38}$ | $15.74_{\pm3.55}$ | $15.35_{\pm3.41}$ | $\mathbf{15.2}_{\pm1.73}$ |
| | R2CCP | $9.86_{\pm1.3}$ | $9.72_{\pm0.49}$ | $10.12_{\pm1.18}$ | $9.7_{\pm1.79}$ | $\mathbf{9.65}_{\pm1.17}$ |
| | SORD | $11.35_{\pm1.65}$ | $13.38_{\pm2.44}$ | $11.42_{\pm2.61}$ | $11.31_{\pm1.35}$ | $\mathbf{11.27}_{\pm1.7}$ |
| | UN | $38.6_{\pm1.89}$ | $43.29_{\pm3.38}$ | $39.89_{\pm1.22}$ | $33.36_{\pm2.12}$ | $\mathbf{28.7}_{\pm1.39}$ |
| Diabetes | DLDL | $35.05_{\pm3.28}$ | $40.13_{\pm2.18}$ | $34.19_{\pm3.02}$ | $33.21_{\pm3.43}$ | $\mathbf{32.81}_{\pm3.58}$ |
| | ELB | $33.3_{\pm3.89}$ | $37.04_{\pm1.71}$ | $31.22_{\pm2.99}$ | $33.18_{\pm2.68}$ | $\mathbf{33.1}_{\pm3.56}$ |
| | R2CCP | $38.16_{\pm4.42}$ | $39.42_{\pm4.97}$ | $32.5_{\pm3.84}$ | $33.76_{\pm3.28}$ | $\mathbf{32.21}_{\pm3.65}$ |
| | SORD | $34.89_{\pm2.99}$ | $22.7_{\pm2.39}$ | $35.03_{\pm4.04}$ | $34.03_{\pm4.32}$ | $\mathbf{33.48}_{\pm3.5}$ |
| | UN | $37.3_{\pm3.68}$ | $39.45_{\pm6.63}$ | $31.91_{\pm4.7}$ | $36.18_{\pm2.92}$ | $\mathbf{35.7}_{\pm3.52}$ |
| Energy | DLDL | $2.88_{\pm0.23}$ | $2.5_{\pm-0.57}$ | $2.9_{\pm-1.36}$ | $2.85_{\pm-1.23}$ | $\mathbf{2.83}_{\pm0.27}$ |
| | ELB | $8.38_{\pm0.26}$ | $8.95_{\pm0.98}$ | $8.17_{\pm0.79}$ | $8.01_{\pm1.62}$ | $\mathbf{7.76}_{\pm0.28}$ |
| | R2CCP | $3.07_{\pm0.26}$ | $2.14_{\pm2.37}$ | $2.96_{\pm1.93}$ | $3.03_{\pm-0.11}$ | $\mathbf{3.02}_{\pm0.29}$ |
| | SORD | $2.95_{\pm0.31}$ | $3.33_{\pm0.15}$ | $3.06_{\pm0.31}$ | $2.94_{\pm1.04}$ | $\mathbf{2.93}_{\pm0.3}$ |
| | UN | $37.6_{\pm0.24}$ | $38.36_{\pm-0.48}$ | $37.16_{\pm-0.68}$ | $28.66_{\pm0.51}$ | $\mathbf{23.8}_{\pm0.28}$ |
| Forest | DLDL | $31.18_{\pm2.45}$ | $23.25_{\pm1.67}$ | $38.13_{\pm2.89}$ | $30.29_{\pm2.35}$ | $\mathbf{29.55}_{\pm2.51}$ |
| | ELB | $27.0_{\pm3.16}$ | $27.31_{\pm3.75}$ | $26.87_{\pm3.08}$ | $26.45_{\pm2.38}$ | $\mathbf{26.2}_{\pm2.57}$ |
| | R2CCP | $30.91_{\pm1.45}$ | $35.76_{\pm1.48}$ | $35.83_{\pm1.88}$ | $28.85_{\pm1.85}$ | $\mathbf{27.79}_{\pm2.5}$ |
| | SORD | $33.57_{\pm3.72}$ | $36.56_{\pm3.61}$ | $33.22_{\pm4.81}$ | $31.02_{\pm3.02}$ | $\mathbf{29.23}_{\pm4.45}$ |
| | UN | $25.4_{\pm3.0}$ | $23.32_{\pm4.86}$ | $27.64_{\pm2.88}$ | $21.47_{\pm3.59}$ | $\mathbf{17.4}_{\pm2.99}$ |
| Parkinsons | DLDL | $2.1_{\pm0.08}$ | $1.56_{\pm0.92}$ | $2.0_{\pm-0.85}$ | $2.09_{\pm0.18}$ | $\mathbf{2.09}_{\pm0.08}$ |
| | ELB | $6.91_{\pm0.06}$ | $7.59_{\pm0.37}$ | $6.78_{\pm-1.11}$ | $6.69_{\pm-0.21}$ | $\mathbf{6.8}_{\pm0.07}$ |
| | R2CCP | $2.12_{\pm0.04}$ | $2.22_{\pm1.29}$ | $2.05_{\pm-0.5}$ | $2.05_{\pm0.21}$ | $\mathbf{2.01}_{\pm0.04}$ |
| | SORD | $2.04_{\pm0.04}$ | $1.83_{\pm0.82}$ | $2.03_{\pm0.91}$ | $2.04_{\pm1.69}$ | $\mathbf{2.04}_{\pm0.04}$ |
| | UN | $33.6_{\pm0.06}$ | $34.85_{\pm-0.61}$ | $32.09_{\pm-1.23}$ | $27.37_{\pm-0.03}$ | $\mathbf{20.8}_{\pm0.07}$ |
| Pendulum | DLDL | $7.15_{\pm1.12}$ | $6.04_{\pm0.39}$ | $6.68_{\pm1.02}$ | $6.77_{\pm1.82}$ | $\mathbf{6.38}_{\pm1.06}$ |
| | ELB | $12.9_{\pm1.27}$ | $14.51_{\pm0.71}$ | $12.39_{\pm0.63}$ | $12.77_{\pm1.39}$ | $\mathbf{12.7}_{\pm0.8}$ |
| | R2CCP | $6.67_{\pm0.95}$ | $7.15_{\pm0.76}$ | $9.06_{\pm0.45}$ | $6.21_{\pm-1.02}$ | $\mathbf{5.86}_{\pm0.77}$ |
| | SORD | $9.23_{\pm1.35}$ | $10.27_{\pm1.43}$ | $9.43_{\pm-0.05}$ | $8.37_{\pm-0.75}$ | $\mathbf{8.08}_{\pm1.22}$ |
| | UN | $16.3_{\pm1.3}$ | $13.04_{\pm-0.19}$ | $17.71_{\pm1.23}$ | $14.83_{\pm2.11}$ | $\mathbf{14.5}_{\pm1.22}$ |
| Solar | DLDL | $22.22_{\pm2.81}$ | $24.47_{\pm2.42}$ | $19.4_{\pm2.36}$ | $20.97_{\pm2.68}$ | $\mathbf{20.03}_{\pm3.17}$ |
| | ELB | $25.1_{\pm3.91}$ | $21.17_{\pm2.86}$ | $26.02_{\pm3.65}$ | $21.54_{\pm3.42}$ | $\mathbf{18.3}_{\pm3.55}$ |
| | R2CCP | $6.22_{\pm0.78}$ | $6.51_{\pm-0.95}$ | $26.99_{\pm1.19}$ | $3.66_{\pm1.09}$ | $\mathbf{1.8}_{\pm0.68}$ |
| | SORD | $21.98_{\pm6.23}$ | $25.38_{\pm5.69}$ | $21.24_{\pm4.59}$ | $18.22_{\pm6.3}$ | $\mathbf{17.53}_{\pm6.08}$ |
| | UN | $11.6_{\pm1.86}$ | $12.03_{\pm5.88}$ | $29.34_{\pm6.03}$ | $9.98_{\pm4.75}$ | $\mathbf{9.29}_{\pm5.96}$ |
| Stock | DLDL | $9.14_{\pm1.04}$ | $9.83_{\pm-0.36}$ | $9.25_{\pm1.25}$ | $8.65_{\pm-0.21}$ | $\mathbf{8.37}_{\pm0.95}$ |
| | ELB | $11.4_{\pm0.91}$ | $10.57_{\pm0.79}$ | $11.07_{\pm0.11}$ | $11.34_{\pm-0.42}$ | $\mathbf{11.3}_{\pm1.03}$ |
| | R2CCP | $9.74_{\pm1.15}$ | $7.88_{\pm1.14}$ | $9.42_{\pm1.68}$ | $9.01_{\pm2.42}$ | $\mathbf{8.86}_{\pm1.25}$ |
| | SORD | $9.73_{\pm0.83}$ | $9.32_{\pm1.03}$ | $9.83_{\pm-1.55}$ | $9.34_{\pm-0.45}$ | $\mathbf{9.25}_{\pm0.98}$ |
| | UN | $25.9_{\pm0.87}$ | $30.41_{\pm1.77}$ | $25.14_{\pm1.25}$ | $24.94_{\pm0.55}$ | $\mathbf{23.9}_{\pm0.98}$ |

### B.3 Additional Experiment Results

In this section, we provide all experiment results of how the sizes of the prediction set change in the calibration and testing sets as we do stepwise posterior alignment.

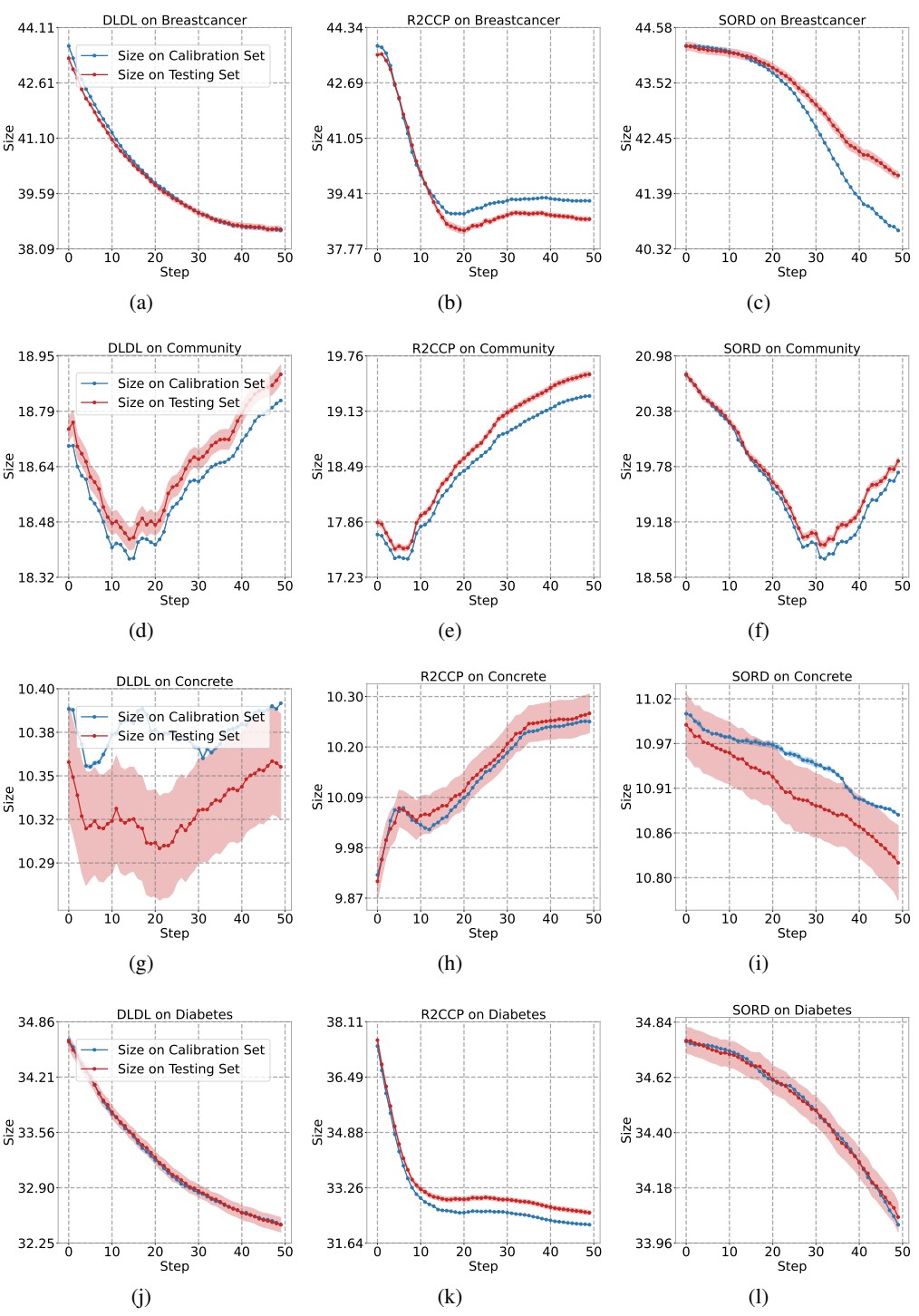

Figure 4: The prediction set size change on Breastcancer, Community, Concrete, and Diabetes

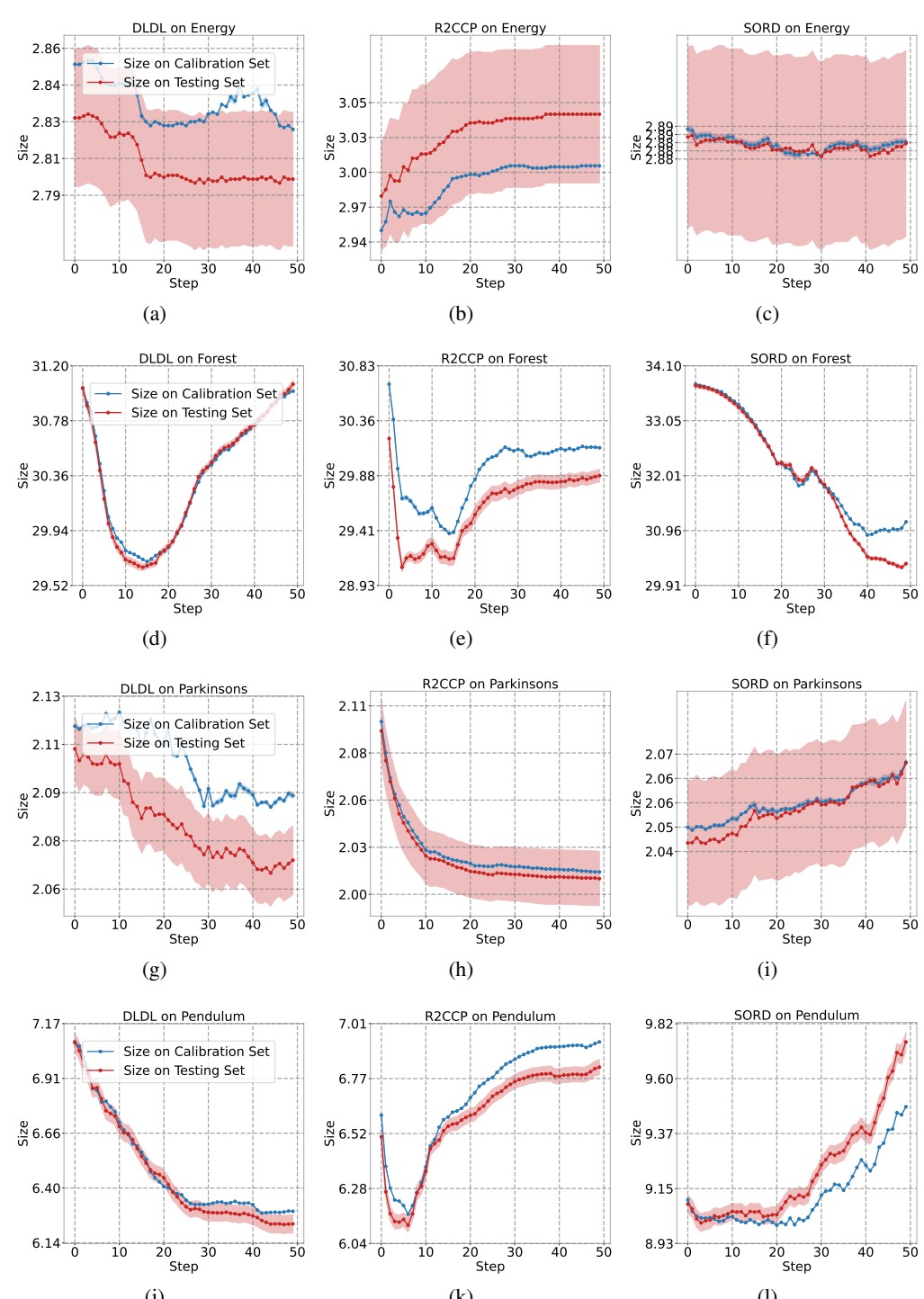

Figure 5: The prediction set size change on Energy, Forest, Parkinsons, and Pendulum

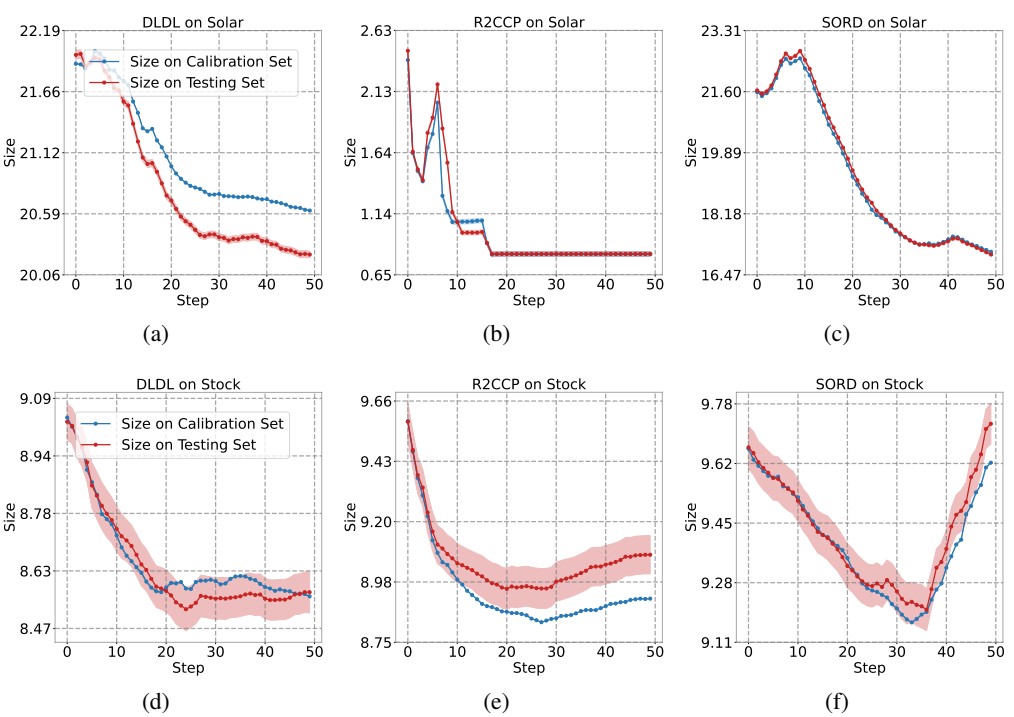

Figure 6: The prediction set size change on Solar and Stock

