# OpenReview forum: "Improving Ordinal Conformal Prediction by Stepwise Adaptive Posterior Alignment"
_ICLR.cc/2025/Conference — ICLR 2025 Conference Withdrawn Submission_

### Official Review · Reviewer_UzjM · 2024-10-20

**Soundness:** 2
**Presentation:** 2
**Contribution:** 2
**Rating:** 3
**Confidence:** 3

**Summary:**

This paper examines conformal prediction for ordinal categorical data. It is widely recognized that setting an appropriate prediction range is an important issue, and that calibrating the estimated probability distribution appropriately is the key to solving the problem. Therefore, it can be said that the paper addresses an important issue for many readers. The focus on the variance of ordinal categorical data is considered to have a certain degree of usefulness, but it is not particularly novel. Regarding the theoretical results (Theorem 1), the assumptions are not necessarily realistic, and I do not think that it provides a theoretical guarantee for the output of the algorithm. Although the discussion is not that mathematically complex, the readability is not very high because the symbols are not always defined appropriately. Overall, after reviewing the paper, I judge that it needs extensive revision in terms of novelty, theory, and experiments in order to be accepted for publication.

**Strengths:**

An algorithm that is relatively easy to implement has been proposed for the purpose of ordinal conformal prediction. The results of Theorem 1 are described as a theoretical consideration. In the numerical experiments, good results were obtained compared to the baseline and competitors. A quantitative study has been conducted on computational cost, etc.

**Weaknesses:**

- Theorem 1 is a theoretical evaluation under a very simple setting, and it does not seem to provide a theoretical guarantee for algorithms with temperature parameters such as the proposed algorithm. What kind of implications does Theorem 1 give for the results of algorithms? Is the distribution learned in such a way that Equation (7) is satisfied by the algorithm?

- Assumption 4 seems to be a strong assumption. Isn't it reasonable to think that the probability of each class always falls within the range of a certain value and twice that value is not true in real data?  Can the authors clearly mention whether this assumption is satisfied in the data used in the numerical experiment?

- In selecting hyperparameters, if there is any noteworthy relationship between hat{s} and q, please specify it.

- The notation is unclear. I can imagine the definitions of the symbols in (10) and (11), but they are difficult to understand. I think it is essential to clearly define and use symbols in academic papers.

**Questions:**

See strengths and weaknesses.

---

### Official Review · Reviewer_JkJw · 2024-11-02

**Soundness:** 3
**Presentation:** 3
**Contribution:** 2
**Rating:** 5
**Confidence:** 4

**Summary:**

Ordinal classification (OC) is used to categorize instances into ordered classes, and in risk-sensitive contexts, ordinal conformal prediction (OCP) generates prediction sets with coverage guarantees. However, OC models often misestimate the posterior distribution, affecting OCP effectiveness. The Adaptive Posterior Alignment Step-by-Step (APASS) method is introduced as a flexible solution that enhances OC models before OCP by reducing distribution discrepancies. APASS uses an attention-based estimator to adaptively estimate posterior variance and aligns it through stepwise temperature scaling. Evaluations show that APASS consistently improves OCP performance across multiple OC models and datasets.

**Strengths:**

- The paper is well-written.
- The proposed method appears novel.
- Several experiments are conducted to support the proposed method.

**Weaknesses:**

- Please provide a complexity analysis of the proposed method.
- Please include a computational comparison with the baselines.

**Questions:**

Please see Weaknesses

---

### Official Review · Reviewer_UqFx · 2024-11-03

**Soundness:** 2
**Presentation:** 3
**Contribution:** 2
**Rating:** 5
**Confidence:** 4

**Summary:**

The paper introduces APASS (Adaptive Posterior Alignment Step-by-Step), a technique to enhance Ordinal Conformal Prediction (OCP) by stepwise aligning posterior variance, aiming to improve prediction efficiency in ordinal classification models by addressing variance misalignment issues.

**Strengths:**

•	Stepwise Innovation: APASS introduces a unique stepwise alignment method that corrects variance misalignment in OCP, resulting in reduced prediction set sizes and improved efficiency.
•	Plug-and-Play Flexibility: APASS is designed as a plug-and-play component, easily integrable with various ordinal classification (OC) models and OCP techniques, enhancing its versatility.

**Weaknesses:**

1.  I find the variance estimation element not sufficiently justified. Furthermore, I recommend that the writers add an experiment of the variance estimator model on some high-dimension synthetic data (where we know the ground-truth variance) to view its performance against some other calibrated OC models. This experiment can demonstrate the necessity and quality of the technique, and if this estimator truly reveals information about the variance that a calibrated OC model does not possess.  Overall, it seems that this method performs more or less calibration for the OC model predictions, so it’s unclear why this method would help OPC to achieve better results, against some other calibrated OC model.

2. APASS requires additional calibration data, which may impact performance in data-limited contexts.'

3. The architecture of the variance estimator model and the rationale behind the chosen loss function are not clearly explained. A more detailed description of the architecture, along with justifications for the design choices and loss function selection, would provide greater clarity and allow for a better understanding of how the variance estimation is achieved.

**Questions:**

None

---

### Official Review · Reviewer_74DW · 2024-11-03

**Soundness:** 3
**Presentation:** 3
**Contribution:** 3
**Rating:** 5
**Confidence:** 3

**Summary:**

In this paper, a method called APASS is proposed as an approach to improve ordinal conformal prediction, a type of uncertainty quantification in the ordinal regression problem. Based on experimental observations that predicted variance can underestimate or overestimate in cases of heteroscedastic noise, the proposed method aims to tighten the conformal prediction set by circumventing this variance misalignment. The theoretical contribution of this paper lies in demonstrating that by reducing the misalignment indicator of posterior variance (Equation (7)), the size of the conformal prediction set can be reduced to the oracle one (Theorem 1). Building on this theoretical finding, a method is proposed to reduce the discrepancy between the predicted posterior variance and the posterior variance of the calibration set by minimizing the weighted sum of prediction errors derived from calibration data.

**Strengths:**

1) Uncertainty quantification with theoretical guarantees is important in the machine learning community. This study holds significance as one of such efforts.

2) The basic notions behind this study—that variance estimation might be biased for data with heteroscedastic noise and that variance correction could produce a more precise conformal prediction set—are reasonable and worth pursuing.

3) The paper is clearly written, allowing even readers unfamiliar with ordinal regression or conformal prediction to understand the significance of the study.

**Weaknesses:**

1) The proposed method is a rather simple extension of an existing approach called Ordinal Conformal Prediction (OCP). There remains concern about whether this extension has sufficient novelty to be accepted at a top-tier conference like ICLR.

2) It is not clearly explained how determining weights using Algorithm 1 to estimate the posterior variance can ensure a reduction in $\eta$ in Equation (7). Without this guarantee, the theoretical consideration of Theorem 1 loses its significance; therefore, if this guarantee exists, it should be properly stated as a theoretical statement.

3) Regarding the determination of the hyperparameter $q$ in Equation (12), which is the main contribution of this paper, there seems to be a possibility of overfitting to the calibration data. It should be clearly explained that using calibration data for variance estimation still preserves the coverage guarantee of the conformal prediction set.

**Questions:**

1) Related to the first weakness mentioned above, could you provide a clear explanation as to why the proposed stepwise update $q$, which constitutes the originality of this research, is indispensable?

2) Related to the second weakness mentioned above, could you provide a clear explanation as to whether Algorithm 1 guarantees a reduction in the misalignment $\eta$ defined in Equation (7)?

3) Related to the third weakness mentioned above, could you provide a clear explanation as to whether the use of calibration data in APASS has no effect on the coverage guarantee of the conformal prediction set?

---

### Official Review · Reviewer_DqVo · 2024-11-03

**Soundness:** 2
**Presentation:** 2
**Contribution:** 2
**Rating:** 3
**Confidence:** 4

**Summary:**

For the ordinal classification problem, the paper highlights the effect of failing to model the posterior distribution accurately on Oridnal-APS (ordinal conformal prediction (OCP)). The paper assumes that the underlying label distribution is uni-modal throughout the entire work. The first result of the work is providing an upper bound of the size/cardinality of the prediciton set produced by OCP, the upper bound is determined by the cardinality of prediction generated with oracle estimator and the discrepancy between the estimated and actual conditional label distributions.

Next, they propose measuring the discrepancy by constructing a posterior variance estimator fitted on the calibration set and using this variance estimator with the posterior variance predicted by the ordinal classification model to perform temperature scaling. Based on empirical evaluations, they propose to do this temperature scaling stepwise; the number of steps is determined by evaluating the effect on the prediction set cardinality on the calibration set.

**Strengths:**

- **Theoretical Justification**: The authors provide a theoretical analysis that explains how reducing misalignment leads to tighter bounds on PS size.
- **Novel Approach:** The introduction of Adaptive Posterior Alignment Step-by-Step, leveraging temperature scaling, posterior variance estimation, and ordinal-APS, seems empirically to be an effective approach resulting in more efficient prediction sets.

**Weaknesses:**

- **Failing to give coverage guarantees:** The approach, while empirically validated, lacks theoretical foundations, which is crucial for conformal prediction approaches. In my opinion, this is impossible to guarantee with your current approach since you use your calibration set for modeling the posterior variance.
- **Failing to give proper credit Dey et al. (2023):** Dey et al. was the first work to stress the importance of unimodal distributions for constructing prediciton sets using conformal prediction. Additionally, they have already given an upper bound of the cardinality of the APS against the oracle set, which is only defined slightly differently, but the philosophy is quite the same. However, I like your presentation more, but you still fail to mention this.
    - Additionally, you say Dey et al. (2023) use a special neural network. However, it is just a final preprocessing layer (like the softmax layer) you must apply.
- **Incomplete benchmark methods:** You should have benchmarked against the technique of Dey et al. (2023). Additionally, the naive prediction set approach, adding the probability until you reach the confidence level, which is similar to how you construct the oracle prediction set, should be used as a benchmark.
- **Inconsistency between Appendix A and the main paper:** I see little consistency between this appendix and the main paper. In Lemma 3, conformity scores and the quantile function are suddenly introduced, which you have not mentioned before.

**Questions:**

- Line 62: sentence grammatically incorrect “*2) a stepwise alignment algorithm that optimizes the calibrate the variance misalignment.”*
- Line 108 should probably mention some of the impossibility results in CP.
- Line 368 typo “sstepwisemethod”

---

### Official Review · Reviewer_2eXa · 2024-11-03

**Soundness:** 3
**Presentation:** 1
**Contribution:** 2
**Rating:** 5
**Confidence:** 4

**Summary:**

The authors propose to adaptively scale the OC method to align the variance in order to improve conformal prediction. The idea is interesting, but the implementation has flaws. The notations need to be clearly defined to enhance the presentation.

**Strengths:**

The alignment of variances to improve the conformal prediction is useful.

**Weaknesses:**

The presentation, especially the notations, needs improvement.

**Questions:**

1: On page 2, line 99, the projection C is defined to map X to 2^Y, which indicates the possible labels of the point in kth class by way of 0,1. How your notation can interpret Y_n+1 \in C_n(X_n+1)?
2. On page 3, in (3) the oracle prediction set is defined using (l, u) \in R^2 (the real values), what's the definition of p(y^j|x) for j=l to u? Note that based on the definition of C function which mapping to 2^y, C(x) is a K-dimensional vector, it is not clear how c(x)=[l,u].
3. In(6), what are the notations m, M B_m? the only nation we know from the formula is n, the number of data points in the calibration set.
4. The assumption 3 on page 5 require the \eta consistency of the estimator of F function. Is this assumption satisfied for any of the OC methods or only some of the methods can produce the consistent estimator?
5. In formula (9), the variance depends on parameter \theta, how does \theta affect the estimation? The index j takes values 1 to K in the first summation, while takes values 1 to N in the second summation. If j is the index for class, the second summation is wrong; if j is the index for points, the first summation is incorrect.
6. Formula (10) has the dependence on \Psi in the variance notation, while on the right side we only have \phi. Whats the relationship of \Psi to \phi in (10) and to \theta in (9)?
7. t(x) is used to scale the prediction to reduce the variance discrepancy, and is applied to scale the prediction f. t(x) is defined to be the ratio of the variances estimated from the calibration set (treated as oracle) and  from the OC estimate, then to making the variance of the scaled estimation to be the same, we just need to take q=1/2 and the scaled f to be f* t(x), not divide t(x) as in the algorithm.

---

### Note · Authors · 2025-01-02

I have read and agree with the venue's withdrawal policy on behalf of myself and my co-authors.